# Isolated Central Nervous System Vasculitides in COVID-19: A Systematic Review of Case Reports and Series

Domizia Vecchio [1,*,†], Francesca Moretto [2,†], Samuel Padelli [1], Francesca Grossi [2], Roberto Cantello [1] and Rosanna Vaschetto [2]

1  Department of Translational Medicine, Section of Neurology, University of Eastern Piedmont, Via Solaroli, 17, 28100 Novara, Italy
2  Department of Translational Medicine, Anesthesia and Intensive Care, University of Eastern Piedmont, Via Solaroli, 17, 28100 Novara, Italy
*  Correspondence: domizia.vecchio@gmail.com
†  These authors contributed equally to this work.

**Abstract:** Cerebral vasculitides, both isolated or in systemic disorders, could be triggered by infections, and few cases have been associated to coronavirus disease 2019 (COVID-19). This study searched for publications in Pubmed, EMBASE, and Cochrane library databases for case reports and series of isolated central nervous system (CNS) vasculitides triggered by severe acute respiratory syndrome coronavirus-2. We included 12 studies (published from June 2020 to July 2022) and collected 39 adult patients (5/39 pathologically or radiologically proven, 34/39 suggestive for primary CNS vasculitis or PCNSV). All cases had a positive real-time polymerase chain reaction on a nasopharyngeal swab or a respiratory tract specimen. About the 85% of the included cases were males, and disease onset occurred later than 50 years old in all but three subjects. In total, 33/39 patients presented severe COVID-19 pneumonia, frequently requiring intensive care unit care. The most common neurological features were headache, obnubilation, and coma. PCNSV was suspected mainly on radiological findings, whereas the cerebrospinal fluid analysis was minimally altered. Magnetic resonance imaging showed vessel wall enhancement in 32/39 cases, generally with the concomitant presence of microbleeds, subarachnoid haemorrhages, and/or multiple ischemic lesions. Despite the severe respiratory and neurological disease course, most cases (93%) improved spontaneously or after a course of high-dose intravenous steroids with no need for immunosuppression. In conclusion, PCNSV could rarely relate to COVID-19 and independently from pulmonary disease severity. Adults with COVID-19-related PCNSV could have a favourable prognosis.

**Keywords:** isolated cerebral vasculitides; primary central nervous system vasculitis; PCNSV; COVID-19; SARS-CoV-2





## 1. Introduction

Cerebral vasculitides is an uncommon disease (2.4 cases per 106 person-years) [1], characterised by inflammation and destruction of the brain blood vessels, that could occur isolated (called primary central nervous system vasculitis or PCNSV), or in systemic disorders. The presentation is variable, ranging from a hyperacute onset, as the occurrence of ischaemic/haemorrhagic strokes, to a chronic insidious course with headache or cognitive impairment [2,3]. Viral infections could themselves cause, by direct invasion or by acting as a trigger, an inflammatory cascade with endothelial dysfunction, apoptosis, vasoconstriction, and subsequent ischaemia, tissue oedema, and procoagulant state [4]. For example, PCNSV onset has been related to previous infection with various microorganisms such as varicella zoster virus [5], West Nile virus [6], hepatitis B and C [7], human immunodeficiency virus [8], and herpesviruses [9]. More specifically, a link has been demonstrated with viruses belonging to the coronaviruses family, such as the Middle East respiratory syndrome-related coronavirus (MERS). In fact, a few cases positive for MERS

showed progressive central nervous system (CNS) manifestations including altered mental status from confusion to coma, ataxia, and focal motor deficits. Nevertheless, MERS has never been found either in brain tissue or in the cerebrospinal fluid [10,11]. Nonfocal brain involvement with encephalopathy, agitation, and confusion have been described in coronavirus disease 2019 (COVID-19) respiratory syndrome [12]. The CNS manifestations of severe acute respiratory syndrome coronavirus-2 (SARS-CoV-2) were related both to viral tropism for neurons [13], glial, and endothelial cells, and to a hyperinflammatory state called "cytokine-mediated storm" [14]. Moreover, McGonagle et al. [15] showed that patients with severe COVID-19 pneumonia could develop systemic vasculitis-like lesions.

PCNSV has been described in few reports and therefore, we performed a review of the current literature to summarise clinical features, imaging and laboratory findings, and treatments and outcomes of patients affected by isolated cerebral vasculitides during COVID-19.

## 2. Materials and Methods

Our literature search was systematically and independently performed by two investigators (FM and SP) on Pubmed, EMBASE and Cochrane library databases through several terms: (COVID-19 or SARS-CoV-2) and (vasculitis or vasculitides). Reports were included if published from 1 March 2020 (the first paper included was published on 26 June 2020) to the 31 July 2022, with English language restriction.

We considered as eligible those reports including patients with: (1) a diagnosis of PC-NSV; cerebral involvement in systemic vasculitides or peripheral nervous system damage were excluded; (2) SARS-CoV-2 concomitant infection; we focused on those cases with positive real-time polymerase chain reaction (PCR) on a nasopharyngeal (NF) swab or a respiratory tract specimen; patients with only clinical features of COVID-19 or the sole immunoglobulin response are not included in the main results; (3) age at onset >17 years old. We excluded patients with pre-existing vasculitides/rheumatological disorders, strokes, or other CNS disorders with disease onset before COVID-19 infection.

We included both cases with pathologically proven or radiologically demonstrated vasculitides (called "definite") according to Salvarani et al. [1] and those suggestive for vasculitides (called "possible"). "Radiologically demonstrated" PCNSV indicated a cerebral angiogram was performed and suggestive of vasculitides. "Possible" PCNSV indicated the clinical presentation was compatible with the diagnosis with exclusion of differentials, plus laboratory and/or imaging support for CNS inflammation. Magnetic resonance imaging (MRI) suggestive for vasculitides included: multiple infarctions and intracranial haemorrhages, small gadolinium-enhanced (GAD+) intracranial lesions or meningeal enhancements, if predominantly affecting small vessels, or vessel wall enhancement, if predominantly affecting large vessels [1,16,17].

## 3. Results

*3.1. Literature Research (Figure 1)*

In total, 2308 potential papers were found (2308/2949 excluding redundant titles or not in English language); 2251 were excluded after reviewing the title and abstract. After reviewing the full text of the remaining 57 papers, 39 were excluded because of an onset in childhood (number or N = 5), reviews of other cases (N = 24), cases of vasculitides without or prior to COVID-19 (N = 6), and cases of systemic vasculitides (N = 4). A putative case of vasculitides versus demyelination was discussed and excluded [18]. Twelve papers were finally included with a total of 39 patients [19–30]. Table 1 summarises the included papers. Five papers with "possible" patients with no positive PCR were not included in the analysis (Table 2) [31–35].

**Table 1.** Clinical characteristics of the included 39 cases with positive real-time polymerase chain reaction on nasopharyngeal swab or respiratory tract specimens.

| First Author | Diagnosis | Gender Age | COVID-19 Testing | COVID-19 Onset | COVID-19 Pneumonia/ ICU | Neurologic Features | MRI | CSF | Pathology | Treatments | Outcome: COVID/ Neurological |
|---|---|---|---|---|---|---|---|---|---|---|---|
| Benguerfi | P | M 74 | PCR (NF) + | Fever, cough | +/+ | Coma | Micro and subarachnoid haemorrhages, multiple ischemic lesions | Normal with SARS-CoV-2 RNA − | No | Steroids * | Recovery |
| Chua | P | F 39 | Respiratory tract + | Fever | −/− | Postpartum headache | Micro and subarachnoid haemorrhages with intracranial arterial narrowing | NA | No | No | Recovery |
| De Oliveira | P | M 69 | PCR (NF) + | Fever, abdominal and chest pain | −/− | Headache, bilateral 4th cranial nerve | Basilar and vertebral arteries (walls GAD+) | Normal cells, mildly increased proteins | No | Steroids * | Recovery |
| Dixon | P | M 64 | PCR (NF) + | Cough, fever | +/+ | Impaired consciousness | Multi-infarcts in bilateral MCA and PCA territories (wall GAD+) | NA | No | Steroids * | Recovery |
| Hanafi | P | M 65 | PCR (NF) + | Cough, fever | +/+ | Coma | Deep white matter vasculopathy with patchy GAD+ | NA | No | NA | NA |
| Kirschenbaum | D | 2 M 70–79 | Respiratory tract + | Respiratory symptoms | +/+ (N = 1) +/− (N = 1) | Coma (N = 1) None (N = 1) | Multiple microbleeds (N = 1) NA (N = 1) | NA | Autopsy: petechial haemorrhages with endotheliitis | NA | Death |
| Lersy | D§/P | 11 (10 M) 61–79 | PCR (NF) or respiratory tract + | Respiratory symptoms | +/+ (N = 9) | Coma (N = 11), pyramidal syndrome (N = 3) | Micro and subarachnoid haemorrhages, ischemic strokes (wall GAD+) §2: intracranial arterial narrowing | 6: normal cells, proteins mildly increased or not, SARS-CoV-2 RNA − | No | NA | NA |
| Raban | P | F 51 | Respiratory tract + | Sore throat, cough | +/− | Hemiparesis | Unilateral ischemic stroke (wall GAD+) | Normal with SARS-CoV-2 RNA − | No | No | Recovery |
| Rettenmaier | D | F 48 | PCR (NF) + | None | −/− | Aphasia | Bithalamic lesion GAD+ | Mildly increased cells, increased proteins | Biopsy: small vessel vasculitides | Steroids * | Improved |
| Strause | P | M 24 | PCR (NF) + | Sore throat, loss of taste and smell | −/− | Hemiparesis, dysarthria, | Multi-infarcts basal ganglia (wall GAD+) | Normal | No | Steroids * | Recovery |
| Uginet | P | 17/31 cases (2 F/31) 53–77 | PCR (NF) + | Dyspnoea | +/+ | Headache, inattention | Cerebral microbleeds (wall GAD+) | 7 CSF: normal cells, proteins mildly increased or not, SARS-CoV-2 RNA − | No | No | Recovery |
| Vaschetto | P | M 64 | PCR (NF) + | Cough, fever | +/+ | Coma, tetraplegia | Multi-infarcts parietal-occipital and pons, leptomeningeal GAD+ | Normal cells, increased proteins | No | IVIG, steroids * | Recovery |

We included both cases with pathologically proven and radiologically demonstrated vasculitides (called "definite") and those suggestive for vasculitides (called "possible"). D: definite vasculitides (N = 5), P: possible vasculitides (N = 34). Other abbreviations: COVID-19: coronavirus disease 2019, CSF: cerebrospinal fluid, DWI: diffusion weighted imaging, EEG: electroencephalogram, F: female, GAD: gadolinium, ICU: intensive care unit, IVIG: intravenous immunoglobulin, M: male, MCA: middle cerebral artery, MRI: magnetic resonance imaging, N: number, NA: not available, PCA: posterior cerebral artery, PCR NF: real-time polymerase chain reaction on nasopharyngeal (NF) swab, RNA: ribonucleic acid, SARS-CoV-2: severe acute respiratory syndrome coronavirus-2. * IV high-dose steroids: methylprednisolone 500–1000 mg daily for 5 days or prednisone 60 mg daily followed or not by tapering.

**Table 2.** Clinical characteristics of the 9 cases with no real-time polymerase chain reaction positive for severe acute respiratory syndrome coronavirus-2.

| First Author | Diagnosis | Gender Age | COVID-19 Testing | COVID-19 Onset | COVID-19 Pneumonia/ ICU | Neurologic Features | MRI | CSF | Pathology | Treatments | Outcome: COVID/ Neurological |
|---|---|---|---|---|---|---|---|---|---|---|---|
| De Sousa | P | M 28 | IgM + | Neurological | −/− | Headache, dysarthria, left hemiparesis | Unilateral parietal and frontal lesions DWI+ | NA | No | NA | NA |
| Ermilov | D | M 20 | NA | NA | NA | NA | NA | NA | Autopsy: widespread vasculitides, thrombosis, haemorrhagic necrosis | NA | Death |
| Lersy | P | M 46 | NA | Respiratory symptoms | +/+ | Delirium | Meningeal and diffuse leptomeningeal inflammation (wall GAD+) | Normal | No | No | Recovery |
| Pugin | P | 3 M, 2 F 69–78 | NA | Fever, dyspnoea | +/+ | Coma | (Wall GAD+) | Normal | No | Steroids * | Recovery |
| Timmons | D | F 26 | NA | Sore throat, taste/smell loss | −/− | Foot drop | Unilateral frontoparietal white matter lesions GAD+ | Normal | Biopsy: lymphocytic vasculitides | Steroids *, my-cophenolate mofetil | Improved |

We included both cases with pathologically proven and radiologically demonstrated vasculitides (called "definite") and those suggestive for vasculitides ("called possible"). D: definite vasculitides (N = 2), P: possible vasculitides (N = 7). Other abbreviations: COVID-19: coronavirus disease 2019, CSF: cerebrospinal fluid, DWI: diffusion weighted imaging, EEG: electroencephalogram, F: female, GAD: gadolinium, ICU: intensive care unit, IgM: immunoglobulin M, M: male, MRI: magnetic resonance imaging, N: number, NA: not available, SARS-CoV-2: severe acute respiratory syndrome coronavirus-2. * IV high-dose steroids: methylprednisolone 500–1000 mg daily for 5 days or prednisone 60 mg daily followed or not by tapering.

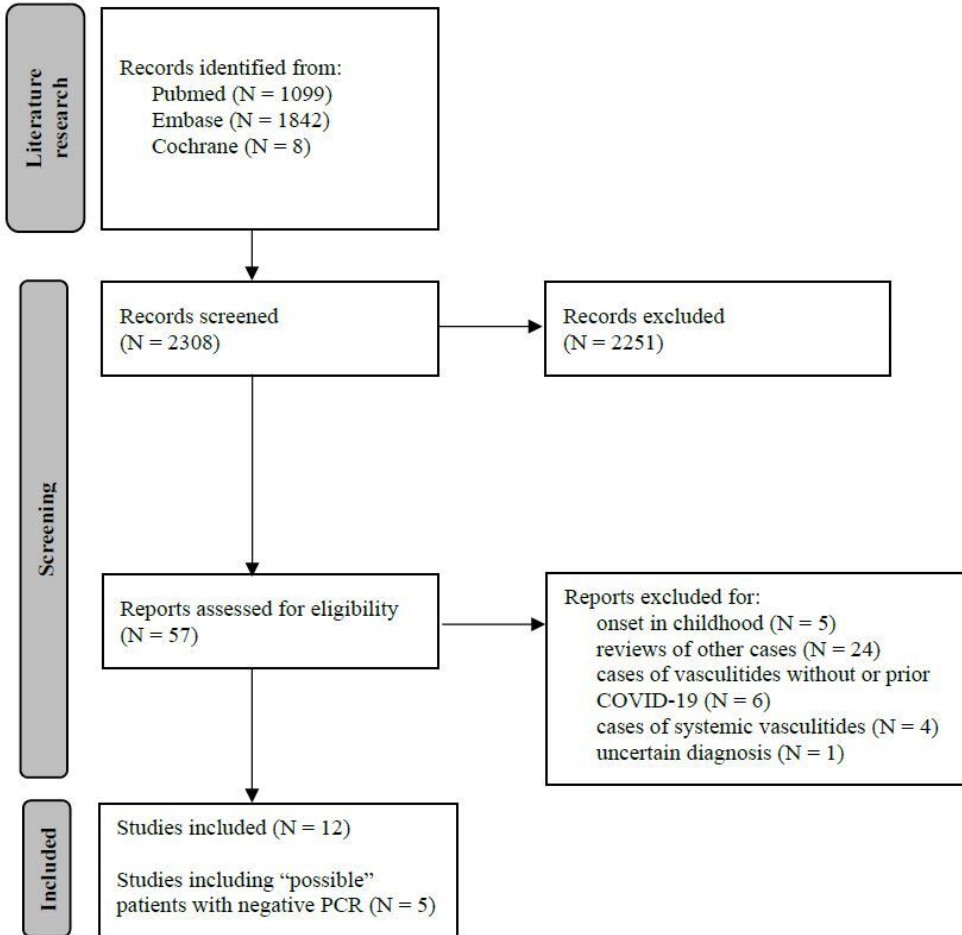

**Figure 1.** Literature research flow chart. Legend: COVID-19: coronavirus disease 2019, PCR: polymerase chain reaction.

### 3.2. Patient Results

Overall, we collected 39 cases of isolated cerebral vasculitides with disease onset during COVID-19. Five patients were classified as "definite" since pathologically proven (three out of five patients, two by autopsy) or radiologically demonstrated (two out of five). The remaining 34 subjects were classified as suggestive for PCNSV (called "possible" vasculitides, according to the clinical and radiological data. About 85% of the included cases were males, and all but three had disease onset over the age of 50 years old. Typical COVID-19 symptoms, including fever, respiratory distress, cough, or abdominal pain, were present in all patients at disease onset except for a 48-year-old female with a neurological onset [27]. SARS-CoV-2 respiratory syndrome was severe in 33 patients that had a pneumonia and mostly required intensive care unit admission. Concerning neurological presentations, about 93% of the patients displayed headache, obnubilation, impaired consciousness, or wakefulness after sedation. Only three patients (8%) showed focal neurological symptoms (hemiparesis or language deficit), none had nausea or vomiting during the clinical course.

MRI was relevant in all cases (not available in a single autoptic case) and supported the diagnosis of cerebral vasculitides. In fact, the most common finding was a vessel wall enhancement in 32/39 patients. Nevertheless, when major cerebral arteries were involved, both the anterior and posterior circulation could be affected. Among those patients with no vessel wall enhancement, leptomeningeal (one case) or lesion patchy enhancement (one case) was described. Microbleeds and subarachnoid haemorrhages with concomitant multiple ischemic lesions supported the hypothesis of small-vessel vasculitis damage in two cases with no GAD+, one by Benguerfi et al. [19] and one confirmed by MR angiogra-

phy in a postpartum 39-year-old female [20]. The remaining three cases were pathologically proven. On the other hand, despite the frequent presence of radiological signs of inflammation, cerebrospinal fluid (CSF) analysis, when available, showed unspecific mild changes, such as minimally increased protein level. In no case SARS-CoV-2 RNA was detected in the CSF.

Disease outcome was available for 27 patients, of whom two died from systemic comorbidities (multiorgan failure, pulmonary hypertension). The remaining 25 cases (93%) improved significantly or recovered completely from PCNSV, despite the severe disease presentation. Six (24%) patients were treated with steroids, given at high dosage, in association to intravenous immunoglobulins in a single case [30]. Nineteen (76%) patients improved and recovered spontaneously. None of our cohort started immunosuppression.

## 4. Discussion

This cohort of 39 adult patients (5/39 pathologically proven or radiologically demonstrated, 34/39 suggestive for PCNSV) presented isolated cerebral vasculitides during COVID-19 that was documented by PCR on respiratory specimens. Compared to PCNSV unrelated to SARS-CoV-2 [36], the selected cases were more frequently males (versus 53% females) and older (versus median onset age of 46 years). Headache and impaired consciousness were the most frequent onset symptoms, considering cerebral vasculitides related or not to COVID-19. Focal neurological deficit was less common in both conditions [37].

If a definite diagnosis could not be confirmed by biopsy or angiogram, an abnormal brain MRI with GAD was the clue for suspecting PCNSV. In fact, in this cohort, the brain scans resembled that of cerebral vasculitides unrelated to COVID-19, showing microbleeds and infarcts with patchy enhancement or major vessel walls with GAD+ [17]. In terms of radiological differentials, we could not identify a predominant lesion location (data not shown). Due to similarities in clinical and radiological features, a haematological disorder (i.e., lymphoma) could be excluded especially in young stroke-like patients [38]. On the contrary, a bilateral asymmetrical periventricular involvement with no signs of bleeding could be more suggestive of demyelination [18].

In this context, CSF analysis resulted only modestly altered in PCNSV unrelated to COVID-19 as in this cohort. No case showed viral RNA in the CSF, similarly to previous reports on MERS [10]. Some authors debated that this finding cannot exclude a virus-related aetiology of the vasculitides, since SARS-CoVs are mainly intraneuronal [39].

Treatment recommendations for PCNSV are mainly based on retrospective studies and current therapeutic regimens are adapted from those validated in systemic vasculitides [40]. In two large cohorts, a favourable response was observed with a high dose of steroids, sometimes in combination with cyclophosphamide [2,41]. Less is known about the treatment of COVID-19-related cerebral vasculitides. In this cohort, most patients were not specifically treated for PCNSV with favourable outcomes. Those cases who underwent steroids had significant improvement too.

Our review has some limitations. Regarding research process, we limited our analysis to three databases (Pubmed, EMBASE, and Cochrane library) and applied an English language restriction. According to the inclusion criteria, only a few cases were "definite" vasculitides. When selecting reports, we double-checked "possible" patients, independently from formal diagnosis in the included paper, for the suspicion of PCNSV according to the clinical presentation, the "vasculitis-like pattern" on MRI, the CSF analysis, and the exclusion of other causes. We did not include those cases that were suggestive for strokes or demyelinating diseases. Furthermore, we limited our research to isolated cerebral vasculitides with no evidence of systemic disease. Being a clinical rarity in adult patients, we did not differentiate among PCNSV with predominant small-vessel or large/medium-vessel involvement [41]. Evidence is mostly limited to isolated reports with no histopathological confirmation, and cerebral angiogram could be recommended in suspected cases. Finally, there is a lack of previous research studies on the topic (according to the recent

pandemic). Other future perspectives focus on searching for any trigger effect of COVID-19 vaccinations for PCNSV and including a paediatric cohort.

## 5. Conclusions

Isolated cerebral vasculitides could be related to COVID-19 and occurred not only in severe forms of SARS-CoV-2 respiratory syndrome. Older males were more affected, and the neurological onset ranged from headache to impaired consciousness, with focal defects less frequently described at presentation. Brain imaging showed multiple ischemic and haemorrhagic lesions with parenchymal/leptomeninges or vessel wall enhancement (thus remaining the pathology or the results of cerebral angiogram mandatory for a "definite" diagnosis). Nevertheless, disease outcome could be favourable, and treatment with steroids could impact on the prognosis with significant improvement.

**Author Contributions:** R.V. and D.V. designed the study. F.M. and S.P. searched the literature and collected the data. D.V. and F.M. wrote the manuscript draft and R.V., F.G. and R.C. revised it. All authors have read and agreed to the published version of the manuscript.

**Funding:** This research received no external funding.

**Institutional Review Board Statement:** The study was conducted in accordance with the Declaration of Helsinki, and according to local Institutional Ethics Committee approval is not needed for reviews.

**Informed Consent Statement:** Not applicable.

**Data Availability Statement:** Not applicable.

**Acknowledgments:** Graphical abstract created with BioRender.com (accessed on 12 August 2022).

**Conflicts of Interest:** The authors declare no conflict of interest.

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
