# Peer review of "Isolated Central Nervous System Vasculitides in COVID-19: A Systematic Review of Case Reports and Series"

_reports, doi:10.3390/reports5030036_

Round 1
Reviewer 1 Report
The authors performed a systematic review (with English language restriction) of the current literature (case reports and series) to summarise clinical features, imaging and laboratory findings, treatments and outcomes of patients affected by isolated cerebral vasculitides during COVID-19.
Twelve papers were included in the review with a total of 39 patients of isolated cerebral vasculitides with disease onset during COVID-19 (5/39 pathologically proven or radiologically demostrated, 34/39 suggestive for primary central nervous system vasculitis).
The authors found that Isolated cerebral vasculitis could be related to COVID-19, and occurred not only in severe forms of SARS-CoV-2 respiratory syndrome. Brain imaging usually showed multiple ischemic and hemorrhagic lesions with parenchymal/leptomeninges or vessel wall enhancement. Nevertheless, disease outcome could be favorable, and treatment with steroids could impact on prognosis with significant improvement.
The study is potentially interesting, but can be improved if the following minor considerations are addressed:
1. Did the isolated central nervous system vasculitides-COVID 19 patients present with nausea or vomiting at any time during the clinical course?
2. The authors adequately point out that magnetic resonance imaging suggestive for vasculitides included: multiple small infarctions and intracranial haemorrhages. However it would be interesting to include in the text a comment on the fact that this neuroimaging scenario may also be caused by hematological diseases. This is a noteworthy aspect that should be emphasized (Expert Review of Hematology 2016; (9), 891-901). Add and comment on the reference.
3. I suggest you include more considerations about the limitation of your study (example: English language restriction).
Author Response
Dear Editor and Reviewers, Thank you very much for your comments. They have been fully addressed in the manuscript (where changes are marked as revisions in red), and here you could find a point-to-point reply.
Reviewer 1
1. Did the isolated central nervous system vasculitides-COVID 19 patients present with nausea or vomiting at any time during the clinical course?
Thank you for specifying. We added: “…none had nausea or vomiting during the clinical course.”
2. The authors adequately point out that magnetic resonance imaging suggestive for vasculitides included: multiple small infarctions and intracranial haemorrhages. However, it would be interesting to include in the text a comment on the fact that this neuroimaging scenario may also be caused by hematological diseases. This is a noteworthy aspect that should be emphasized (Expert Review of Hematology 2016; (9), 891-901). Add and comment on the reference.
Thank you for the suggestion, we changed the text as:
In terms of radiological differentials, we could not identify a predominant lesion location (data not shown). Due to similarities in clinical and radiological features, a haematological disorder (i.e. lymphoma) could be excluded especially in young stroke-like patients [38]. On the contrary, a bilateral asymmetrical periventricular involvement with no signs of bleeding could be more suggestive of demyelination [18].
Arboix A, Jiménez C, Massons J, Parra O, Besses C. Hematological disorders: a commonly unrecognized cause of acute stroke. Expert Rev Hematol. 2016 Sep;9(9):891-901
3. I suggest you include more considerations about the limitation of your study (example: English language restriction).
Thank you for the suggestion, we added: English language restriction, the databased selected for the research, and the lack of previous research studies on the topic (according to the COVID-19 pandemic).
Reviewer 2 Report
The authors Vecchio et al., Sum up an overview of Cerebral vasculitides
as consequences of Covid-19.
The literature cited is limited by the specific clinical questions, and recent updates contribute to elucidating a potential experimental work.
The authors try to introduce their own style, interpretation as well as an outlook on this interesting topic. The authors arrange a review that does not hold an individual flavor.
Possible explanations for neurologic diseases are discussed.
I suggest shifting Table 2 from supplemental to the main text because aimed at reader support. In addition to graphical abstract/cartoon on cerebral vasculitides mechanisms
and potential association to COVID-19 is appropriate.
The length of the paper is commensurate with the message.
The review may be accepted for publication in this journal with minor revisions.
Author Response
Fh
Dear Editor and Reviewers, Thank you very much for your comments. They have been fully addressed in the manuscript (where changes are marked as revisions in red), and here you could find a commet.
The authors Vecchio et al., Sum up an overview of Cerebral vasculitides as consequences of Covid-19.
The literature cited is limited by the specific clinical questions, and recent updates contribute to elucidating a potential experimental work.The authors try to introduce their own style, interpretation as well as an outlook on this interesting topic. The authors arrange a review that does not hold an individual flavor. Possible explanations for neurologic diseases are discussed.
I suggest shifting Table 2 from supplemental to the main text because aimed at reader support. In addition to graphical abstract/cartoon on cerebral vasculitides mechanismsand potential association to COVID-19 is appropriate.
The length of the paper is commensurate with the message.The review may be accepted for publication in this journal with minor revisions
Thank you for the suggestion, we moved Table S1 as Table 2 in the main text.
Reviewer 3 Report
Thank you for inviting me to review this manuscript.
In this review the authors searched for publications in different database for case reports and series of isolated central nervous system (CNS) vasculitides triggered by Severe Acute Respiratory 15 Syndrome Coronavirus-2.
I have enjoyed reading this manuscript and I found it very interesting and, even if I’m not an expert on the technical issues related to literature reviews, to me, the work seems well performed and sufficiently detailed.
I only have a suggestion to improve it: What I think is missing is a practical final message aimed at guiding future studies. In the conclusions, I find that in addition to summarizing the results and giving a suggestion of therapy, it is useful to add suggestions for possible future studies.
Author Response
Thank you for the suggestion, we added perspectives at the end of limitations according to the lack of literature in the topic.